# Genetic Variants as Predictive Markers for Ototoxicity and Nephrotoxicity in Patients with Locally Advanced Head and Neck Cancer Treated with Cisplatin-Containing Chemoradiotherapy (The PRONE Study)

**DOI:** 10.3390/cancers11040551

**Published:** 2019-04-17

**Authors:** Chantal M. Driessen, Janneke C. Ham, Maroeska te Loo, Esther van Meerten, Maurits van Lamoen, Marina H. Hakobjan, Robert P. Takes, Winette T. van der Graaf, Johannes H. Kaanders, Marieke J.H. Coenen, Carla M. van Herpen

**Affiliations:** 1Department of Medical Oncology, Radboud University Medical Center, Postbox 9101, 6500 HB Nijmegen, The Netherlands; Chantal.driessen@radboudumc.nl (C.M.D.); Winette.vanderGraaf@radboudumc.nl (W.T.v.d.G.); carla.vanherpen@radboudumc.nl (C.M.v.H.); 2Department of Pediatric Hematology and Oncology, Radboud University Medical Center, Postbox 9101, 6500 HB Nijmegen, The Netherlands; Maroeska.teLoo@radboudumc.nl; 3Department of Medical Oncology, Erasmus MC Cancer Institute, Postbox 2040, 3000 CA Rotterdam, The Netherlands; e.vanmeerten@erasmus.nl; 4Department of Otorhinolaryngology and Head and Neck Surgery, Radboud University Medical Center, Postbox 9101, 6500 HB Nijmegen, The Netherlands; mauritsvanlamoen@hotmail.com (M.v.L.); Robert.takes@radboudumc.nl (R.P.T.); 5Department of Human Genetics, Radboud Institute of Health Sciences, Radboud University Medical Center, Postbox 9101, 6500 HB Nijmegen, The Netherlands; Marina.Hakobjan@radboudumc.nl (M.H.H.); Marieke.coenen@radboudumc.nl (M.J.H.C.); 6Department of Radiation Oncology, Radboud University Medical Center, Postbox 9101, 6500 HB Nijmegen, The Netherlands; j.kaanders@radboudumc.nl

**Keywords:** chemoradiotherapy, cisplatin, ototoxicity, nephrotoxicity, *ACYP2*, genetic variants, SNP

## Abstract

Ototoxicity and nephrotoxicity are potentially irreversible side effects of chemoradiotherapy with cisplatin in locally advanced head and neck cancer (LAHNC) patients. Several predictive genetic variants have been described, but as yet none in LAHNC patients. The aim of this study is to investigate genetic variants as predictors for ototoxicity and nephrotoxicity in LAHNC patients treated with cisplatin-containing chemoradiotherapy. Our prospective cohort of 92 patients was genotyped for 10 genetic variants and evaluated for their association with cisplatin-induced ototoxicity (*ACYP2*, *COMT*, *TPMT* and *WFS1*) and nephrotoxicity (*OCT2*, *MATE* and *XPD*). Ototoxicity was determined by patient-reported complaints as well as tone audiometrical assessments. Nephrotoxicity was defined as a decrease of ≥25% in creatinine clearance during treatment compared to baseline. A significant association was observed between carriership of the A allele for rs1872328 in the *ACYP2* gene and cisplatin-induced clinically determined ototoxicity (*p* = 0.019), and not for ototoxicity measured by tone audiometrical assessments (*p* = 0.449). Carriership of a T allele for rs316019 in the *OCT2* gene was significantly associated with nephrotoxicity at any time during chemoradiotherapy (*p* = 0.022), but not with nephrotoxicity at the end of the chemoradiotherapy. In conclusion, we showed prospectively that in LAHNC patients genetic variants in *ACYP2* are significantly associated with clinically determined ototoxicity. Validation studies are necessary to prove the added value for individualized treatments plans in these patients.

## 1. Introduction

Head and neck cancer is a common type of cancer worldwide [1,2]. The most frequently used treatment for patients with locally advanced head and neck cancer (LAHNC) is concomitant chemoradiotherapy with cisplatin, which improves loco-regional control as well as overall survival compared to radiotherapy alone [3,4]. Chemoradiotherapy can also be applied as adjuvant treatment in case of a high recurrence risk after surgery. Concomitant chemoradiotherapy with cisplatin however, induces a high rate of acute toxicities such as mucositis, dysphagia and dermatitis, most of which will recover with time, but can also induce irreversible ototoxicity and nephrotoxicity [3,5,6,7].

Ototoxicity, characterized by sensorineural hearing loss, can be an adverse effect of either systemically administered cisplatin or radiotherapy to the inner ear. Both chemotherapy and radiotherapy cause lesions in the cochlea, which may lead to ototoxicity [8]. Ototoxicity caused by cisplatin begins with high frequency loss and is often bilateral, permanent and can be progressive also after the end of administration of cisplatin [8,9]. Chemotherapy with cisplatin is most often applied either at a dose of 100 mg/m^2^ every three weeks for three cycles (high dose) or at a dose of 40 mg/m^2^ every week for six or seven cycles (intermediate dose) [5,6]. The incidence of ototoxicity in patients treated with high dose cisplatin is 79% [10]. Besides ototoxicity, another common side effect of cisplatin is nephrotoxicity that also can be irreversible. However, unlike ototoxicity, part of the nephrotoxicity can be reduced by hyperhydration with high natriumchloride. The occurrence and the severity of nephrotoxicity is also related to the cisplatin dose; 100% of the patients treated with high dose cisplatin experienced nephrotoxicity of any grade compared with 75% of the patients with intermediate dose cisplatin [7].

With the aim to prevent ototoxicity and nephrotoxicity, several studies have been performed to identify risk factors and predictive markers. Known clinical risk factors for ototoxicity after chemoradiation in LAHNC patients are the cumulative dose of cisplatin and cumulative radiation dose to the cochlea, younger age, good pretreatment hearing, administration of furosemide and low levels of serum albumin and hemoglobin [11,12]. However, cisplatin-induced toxicity can only partly be predicted by these factors. Recently, various studies found genetic variants, i.e., single nucleotide polymorphisms (SNPs), that are associated with cisplatin-induced side-effects. Genetic variants in acylphosphatase 2 (*ACYP2*) and Wolframin ER transmembrane glycoprotein (*WFS1*) were identified as predictive markers for hearing loss [13,14,15,16,17]. The *ACYP2* gene is expressed in the cochlea [14]. Mutations in *WFS1* can cause progressive deafness after administration of cisplatin [18]. Cisplatin-induced ototoxicity could be related to increased levels of *S*-adenosylmethionine through reduced thiopurine *S*-methyltransferase (*TPMT*) or catechol *O*-methyltransferase (*COMT*) activity. However, their predictive value for ototoxicity is controversial [19,20]. With respect to nephrotoxicity, genetic variants in organic cation transporter 2 (*OCT2*), (multidrug and toxin extrusion 1 (*MATE1*) and xeroderma pigmentosum group D (*XPD*), are believed to be of predictive value [21,22,23,24]. *OCT2* and *MATE1* are expressed in the human kidney at the basolateral membrane of renal proximal tubules, and are involved in the secretion of various cationic substances from the circulation into tubular cells. In that way *OCT2* and *MATE1* are involved in the cellular uptake of cisplatin [21,23]. *XPD* is part of the nucleotide excision repair pathway and is involved in removal of cisplatin and radiotherapy induced DNA damage [24,25].

However, most of the abovementioned studies showed correlations in a limited number of patients and thus confirmation of the association between the SNPs and cisplatin related side effect is needed. Furthermore, the studies were performed in patients with other cancers than LAHNC. Therefore, the aim of this study is to investigate the relationship between the different SNPs and cisplatin-induced ototoxicity and nephrotoxicity in LAHNC patients. 

## 2. Materials and Methods

### 2.1. Patients and Treatment

A cohort of Dutch patients with pathologically proven LAHNC and treated with cisplatin-based chemoradiotherapy was prospectively recruited at the Radboud University Medical Center and the Erasmus University Medical Center in The Netherlands. Eligibility criteria included a minimum age of 18 years and a WHO performance score of 0 or 1. Patients with renal dysfunction defined as a creatinine clearance below 60 mL/min were not considered eligible. Before inclusion, written informed consent was obtained from all patients. The local ethical committee waived the study from ethical approval.

Patients were treated with cisplatin-based chemoradiotherapy for either primary treatment or adjuvant treatment. Concomitant chemoradiotherapy was administered in three different treatment schedules; (1) Conventional radiotherapy in combination with cisplatin 100 mg/m^2^ on days 1, 22 and 43; (2) Accelerated radiotherapy combined with cisplatin 40 mg/m^2^ on a weekly basis for 6 weeks; (3) Conventional radiotherapy combined with cisplatin 40 mg/m^2^ on a weekly basis for 7 weeks. Intensity-modulated radiation therapy (IMRT) was mandatory. Dose to gross tumor volume was 68–70 Gray (Gy) and dose to elective nodal areas 46–50.3 Gy.

Cisplatin was given by infusion in combination with standard prehydration, posthydration and anti-emetics. If during the treatment the creatinine clearance was below 60 mL/min because of dehydration, cisplatin was only administered if the creatinine clearance recovered to 60 mL/min after rehydration. Dose modifications and discontinuation of cisplatin were performed according to standard local practice.

### 2.2. Measurements

Tone audiometry was performed according to standard procedures under standardized conditions. Air-conduction and thresholds were determined at 1, 2, 4, 8, 10, 12.5 and 16 kHz. Bone-conduction thresholds were measured, from 1 kHz to 8 kHz. According to the study protocol, audiometry was carried out at baseline, during chemoradiotherapy after 100 mg/m^2^ or 120 mg/m^2^ cisplatin as total dosage at that moment and within 2 months after completion of treatment. 

Ototoxicity was scored utilizing two different approaches. In the first approach, clinically determined ototoxicity, physicians asked their patients to the hearing loss according to the Common Terminology Criteria for Adverse Events (CTCAE) version 4.03. In the second approach, hearing loss was classified using the tone audiometric data from baseline and end of treatment based on the ear with the worst hearing loss. Hearing loss was defined by threshold shifts at 2, 4 or 8 kHz of ≤25 dB (grade 1), threshold shift of 26–40 dB (grade 2) or threshold shift of ≥40 dB (grade 3). [15]

Additionally, weekly laboratory tests were performed including the creatinine clearance by use of calculation of the Modification of Diet in Renal Disease (MDRD). Nephrotoxicity was defined as a decrease of 25% or more in creatinine clearance by the MDRD at any point during treatment compared to baseline, based on the international accepted Risk, Injury, Failure, Loss and End Stage Renal Disease (RIFLE) criteria [7]. Blood or saliva (Oragene saliva collection kit, DNA Genotek, Kanata, ON, Canada) were used for DNA extraction. 

### 2.3. Genotyping

Genotyping of genetic variant in *TPMT* (rs12201199, rs1800460, rs1142345) and *COMT* (rs9332377) were performed using Taqman SNP genotyping according to the instructions of the manufacturer (ThermoFisher, Nieuwerkerk aan den IJssel, The Netherlands). The other genetic variants (*COMT* rs4646316, *ACYP2* rs1872328, *OCT2/SLC22A2* rs316019, *WFS1* rs62283056, *XPD/ERCC2* rs13181 and *MATE1* rs2289669) were genotyped using Kompetitive Allele Specific PCR (KASP^TM^, KASPar-On_Demand assays (Laboratory of the Government Chemist (LGC) Genomics, Hoddesdon, UK)) according to the instructions of the manufacturer [13,26]. Analysis of the Taqman and KASP assay was carried out on a 7500FAST Real-Time PCR System (Thermo Fisher, Nieuwerkerk aan den IJssel, The Netherlands). Genotypes were scored using 7500 software (v2.0.6, Thermo Fisher). Negative controls as well duplicates (8%) were included as quality controls for genotyping.

### 2.4. Statistics

A sample size calculation showed that inclusion of 100 patients in our study would give 80% power to identify a statistically significant association between a SNP and ototoxicity, assuming a 40% ototoxicity rate, an alpha of 0.05, an allelic odds ratio (OR) of 3, and a minor allele frequency of 10% [19,20].

The association between the SNPs and clinically relevant hearing loss (“yes” vs. “no”) and between the SNPs and nephrotoxicity (“yes” vs. “no”) were analyzed with a Pearson’s chi-square or Fisher-exact tests. *p*-values were tested two-sided and were considered as statistically significant when <0.05. SPSS version 22 (New York, United States) was used for performing the analyses. Meta-analysis of the data of ototoxicity and *ACYP2* was performed using a fixed-effects model in review Manager version 5.3 (The Cochrane Collaboration, Oxford, UK).

## 3. Results

Between August 2013 and February 2017, 103 patients were included in this study. One patient withdrew consent. In 10 cases no blood or saliva samples were available for DNA analysis. Thus, in total 92 patients were included in the final analysis. Fifty-seven patients were treated with intermediate dose cisplatin 40 mg/m^2^ weekly for 6 or 7 weeks, and 35 patients were treated with high dose cisplatin 100 mg/m^2^ on days 1, 22 and 43. Baseline characteristics are shown in Table 1.

### 3.1. Ototoxicity

In all 92 patients, data on clinically-determined ototoxicity were available, whereas hearing loss after treatment based on tone audiometric measurements was available for 79 patients. Of the 92 patients, six patients reported new grade 2 hearing loss and one patient reported grade 3 hearing loss at end of treatment. (Table 2) Of these seven patients, four were treated with cisplatin 40 mg/m^2^ and three were treated with cisplatin 100 mg/m^2^. Based on audiometric measurements, of the 79 patients included in the analysis, 52 patients (65.8%) had grade 1 hearing loss, whereas 16 patients (20.3%) and 11 patients (13.9%) had grade 2 and 3 hearing loss, respectively (Table 2). Nine of the 11 patients with grade 3 hearing loss were treated with cisplatin 100 mg/m^2^. There was no statistically difference in cumulative cisplatin dose in patients with or without hearing loss when measured clinically or audiometrically (*p* = 0.231 and *p* = 0.142). 

Unfortunately, bone conduction was only available in 55 patients. Of these 41 patients (74%) showed no hearing loss, 11 patients (20%) showed mild hearing loss, three patients (6%) suffered moderate-profound hearing loss. (Table 2)

### 3.2. Nephrotoxicity

Of the 92 patients, 53 patients (58%) had nephrotoxicity at any time during treatment. (Table 2) Cumulative cisplatin dose was not different between those patients intentionally treated with high dose or intermediate dose cisplatin (*p* = 0.107). In 86 patients end of treatment creatinine clearance was available. Of these patients, eight (9%) had nephrotoxicity relative to baseline. All eight of these patients (100%) were treated with cisplatin 100 mg/m^2^. 

### 3.3. SNP and Ototoxicity

Nine patients were heterozygous GA for the *ACYP2* variant rs1872328; all other patients (*n* = 83) were homozygous GG. Forty-three percent of the patients reporting clinically hearing loss grade 2 or 3 (3 out of 7 patients) were carrying an A allele, whereas 7% of the patients without clinically hearing loss (grade 0 or 1) were carrier of the A allele. Association analysis showed a statistically significant difference between the groups (*p* = 0.019, OR 9.9, 95% confidence interval (CI) (1.8–54.7)). We found no differences between carriership of the A allele and ototoxicity based on tone audiometrical measurements (Table 3). A meta-analysis of the cohorts of the previous published studies performed in humans also indicated a significant association of the *ACYP2* variant with ototoxicity. For this analysis we used the data of the audiometrical assessments in our patients to compare with the other studies (Figure 1).

For the tested genetic variants in *TPMT*, *COMT* and *WFS1* no statistically differences were found in either clinically or tone audiometrically assessed hearing loss. (Table 3) Also, the association analysis in patients with hearing loss using bone conduction as outcome, showed no statistically significant difference.

### 3.4. SNP and Nephrotoxicity

Data on the *OCT2* gene were available in 93 patients. Eighteen patients were heterozygous GT for the *OCT2* variant rs316019; three patients were homozygous TT and all other 72 patients were homozygous GG. Thirty percent of the patients with nephrotoxicity during treatment were carrying a T allele, whereas 12.5% of the patients without nephrotoxicity, which was significantly different (*p* = 0.049, OR 3.78 95%CI (1.1–12.4)). No association was found between carriers of the T allele and nephrotoxicity at end of treatment compared to baseline (*p =* 0.845). Nephrotoxicity was not significantly associated with the analyzed genetic variants in *MATE1* and *XDP* (Table 4).

## 4. Discussion

Since a high percentage of LAHNC patients treated with cisplatin-based chemoradiotherapy develop irreversible ototoxicity and nephrotoxicity, it would be worthwhile to add predictive biomarkers for toxicity to treatment decision-making to avoid these. In this study we investigated whether germline genetic variants were associated with ototoxicity and nephrotoxicity. We focused on 10 SNPs in seven genes which were previously reported to be related to these adverse effects [15,16,19,21]. We could confirm the association between a genetic variant in *ACYP2* and clinical reported hearing loss, but not with tone audiometrical measurements. Moreover, we found an association with *OCT2* and nephrotoxicity during treatment with cisplatin, but not with nephrotoxicity at end of treatment, which makes it not useful in clinical practice. 

With our findings we are the fifth group to report on the association between genetic variation in the *ACYP2* gene and cisplatin-induced ototoxicity [13,14,15,16]. The initial studies of Xu and Vos reported that the A allele of the genetic variants rs1872328 in the *ACYP2* gene was only present in patients with ototoxicity, i.e., 13.8% and 6.5%, respectively, carried the A allele. More recent studies, also identified the A allele in patients without hearing loss, but only in a low percentage (1%) [15]. In contrast to these studies we found that 57% of the patients without audiometrical measured ototoxicity carried the A allele and 43% of the patients with mild to moderate audiometrical measured ototoxicity. A study by Fang et al. described an increased risk of esophageal carcinoma associated with the genetic variant rs11125529 in the *ACYP2* [27]. Although another variant of the A allele was found and a Chinese population was studied, a genetic variant of the A allele could be related to head and neck cancer and therefore found more often in our cohort.

We could not find an association between the other variants investigated and ototoxicity. This is in line with previous studies that showed equivocal results [15,19,20] (Table 5). A possible confounder in ototoxicity rate in our patient population is radiation in the head and neck region, because radiation can induce conductive hearing loss as a result of inflammation and edema as well as sensorineural hearing loss caused by radiation on the inner ear [11]. Although some patients in the studies by Xu et al. and Ross et al. received cranial radiation as well [14,19].

There is a great variance in the applied scoring systems for ototoxicity between the studies, as the initial studies were done in children, most systems are only validated in children [28]. We decided to perform two analyses, one based on clinical hearing loss and the other on objective audiometrical assessments. For the audiometrical assessments we used the same scoring system as Drogemoller, because this system can be applied to adults, in contrast to the Chang scoring system which is only used for children [13,15]. We are the first to use clinically-determined ototoxicity as well, as this is a clinically relevant outcome measure reflecting the patients’ perspective. However, it remains a subjective outcome, which can be influenced by the interpretation of different investigators. Interestingly we could detect an association between the genetic variant in *ACYP2* when using the clinical measure but not for audiometrically determined ototoxicity. The reason for the discrepancy between clinical and audiometrical determined ototoxicity is speculative. Audiometrical determined ototoxicity is obviously more objective, but clinical assessed ototoxicity is probably more relevant for the patient. In the meta-analysis that we performed, our study had the same direction of effect (OR > 1) as the other studies.

Cisplatin-induced sensorineural hearing loss can best be evaluated with bone conduction measurements as this specifically measures the hearing of the inner ear/cochlea [29]. Theunissen et al. [29] argued that air conduction thresholds represent the functionality of the whole auditory system, including both air and bone conduction, and felt that the grading criteria should comprehend the overall hearing loss due to treatment as this is eventually the clinically relevant hearing loss the patient experiences.

However, to address the underlying mechanism of cisplatin-associated hearing loss and the relation with genetic variants, we decided to perform an association analysis using both air and bone conduction hearing tests, but we did not find any association between the studied genetic variants and the two ototoxicity outcomes. 

With respect to SNPs as predictive markers for nephrotoxicity, the genetic variant in *OCT2* was found to be significant associated with nephrotoxicity at any point during chemoradiotherapy, but not with nephrotoxicity at end of treatment. To our knowledge, only two studies have been performed to assess the relationship between *OCT2* and cisplatin-induced nephrotoxicity in humans. [21,30]. Filipski et al. [21] investigated the effect of the rs316019 variant in *OCT2* in 78 cancer patients receiving cisplatin. Renal function was determined one day before and 1–8 days after the first dose cisplatin. Iwata et al. investigated the rs316019 variant of *OCT2* in 53 patients receiving cisplatin during more cycles. Remarkably, both Iwata et al. and Filipski et al. showed that the presence of T of the genetic variant rs316019 in *OCT2* was ameliorating cisplatin-induced nephrotoxicity, whereas our study found the opposite. The variation in the results might be related to the different endpoints for nephrotoxicity that have been used. Based on the previous studies and ours, we believe that at the moment the use of this SNP is not relevant for clinical practice. 

In our study, patients treated with high dose cisplatin and intermediate dose cisplatin were taken together, because of the small number of patients treated with the high dose schedule. Therefore we cannot draw conclusion regarding association between SNPs and toxicity for specific cisplatin dosages, while we know from previous studies that high dose cisplatin induces higher rates of ototoxicity as well as nephrotoxicity. These data could be of particular interest, as several studies demonstrated the ability to enhance chemosensitivity to cisplatin in HNC, thus reducing the dosage needed, through the down-regulation of molecules related to cell death. [31,32] 

A limitation of our study is that we were not able to reach the planned sample size of 100 patients, due to lack of DNA of 10 patients, resulting in a somewhat smaller patient cohort. Furthermore, because of the relative small patient population, we did not correct for multiple testing and could not perform subgroup analyses. Therefore this study should be viewed as the first steps in the link between the studied genes and toxicities in LAHNC patients.

## 5. Conclusions

This is the first study to the association of ACYP2 and cisplatin-induced ototoxicity in LAHNC patients and the fifth to describe the possible predictive value of ACYP2 regarding (clinical determined) cisplatin-induced ototoxicity. These findings should be validated in a large cohort, to finally determine the predictive value of ACYP2 in ototoxicity. As personalized medicine is getting more important, these findings could eventually provide better tailored, individualized treatment for LAHNC patients considering both oncologic efficacy as well as toxicity and quality of life, as other treatment regimens are available such as radiotherapy with cetuximab or carboplatin.

## Figures and Tables

**Figure 1 cancers-11-00551-f001:**
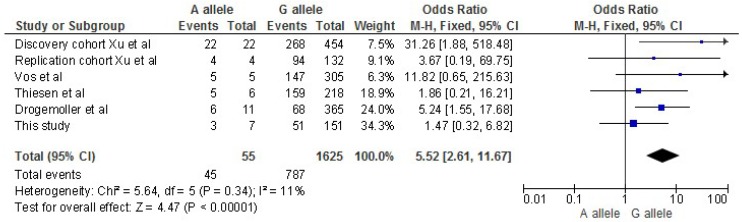
Forest plot of meta-analysis of *ACYP2* rs1872328. Meta-analysis of published cohorts in humans and present study using a fixed-effects model. 95% CI, 95% confidence interval of odds ratio; M-H, Mantel-Haenszel method.

**Table 1 cancers-11-00551-t001:** Patient characteristics of the 92 patients analyzed. WHO: World Health Organisation

Patient Characteristic	Number of Patients (%)
Age mean (range)	57.8 (28–69)
Gender	
**Male**	67 (72.8)
**Female**	25 (27.2)
WHO score	
**0**	63 (68.5)
**1**	28 (30.4)
**2**	1 (1.1)
Treatment indication	
**Primary treatment**	62 (67.4)
**Postoperative treatment**	29 (31.5)
**Primary treatment tumor, postoperative**	1 (1.1)
**Treatment for lymph nodes**	
Primary site	
**Oral cavity**	21 (22.8)
**Oropharynx**	42 (45.7)
**Hypopharynx**	10 (10.9)
**Larynx**	13 (14.1)
**Unknown primary**	4 (4.3)
**Nasal vestibule**	2 (2.2)
Cisplatin dose	
**40 mg/m^2^**	57 (61.3)
**100 mg/m^2^**	35 (37.6)
Cumulative cisplatin dose (median, range)	240 mg (80–300)

**Table 2 cancers-11-00551-t002:** Ototoxicity and nephrotoxicity. MDRD: Modification of Diet in Renal Disease

Toxicity	Number of Patients (%)
Ototoxicity clinically at the end of treatment (*n* = 92)	
None (grade 0 or grade 1)	85 (92.4)
Hearing loss without hearing aid indicated (grade 2)	6 (6.5)
Hearing loss with hearing aid indicated (grade 3)	1 (1.1)
Ototoxicity by tone audiometry (*n* = 79)	
Grade 1 (≤25 dB loss)	52 (65.8)
Grade 2 (26–40 dB loss)	16 (20.3)
Grade 3 (≥40 dB loss)	11 (13.9)
Ototoxicity by audiometry, only bone conduction (*n* = 55)	
Grade 1 (≤25 dB loss)	41 (74.5)
Grade 2 (26–40 dB loss)	11 (20)
Grade 3 (≥40 dB loss)	3 (5.5)
Nephrotoxicity any time during study (*n* = 92)	
MDRD < 25% decrease	39 (42.4)
MDRD ≥ 25% decrease	53 (57.6)
Nephrotoxicity at the end of study (*n* = 86)	
MDRD < 25% decrease	78 (84.8)
MDRD ≥ 25% decrease	8 (8.7)

**Table 3 cancers-11-00551-t003:** Significance levels for genetic variants related to ototoxicity. * = significantly different (*p* < 0.05).

Genotype	Clinical No Ototoxicity N (%)	Clinical Ototoxicity N (%)	*p*-Value	Audiometrical Ototoxicity Grade 1 N (%)	Audiometrical Ototoxicity Grade 2 N (%)	Audiometrical Ototoxicity Grade 3 N (%)	*p*-Value
TPMT:rs12201199							
AA	77 (91.6)	7 (8.3)	*p* = 1.0	47 (66)	14 (20)	10 (14)	*p* = 0.863
AT/TT	8 (100)	0 (0)		5 (63)	2 (25)	1 (12)	
TPMT:rs1142345							
TT	77 (91.6)	7 (8.3)	*p* = 1.0	47 (66)	14 (20)	10 (14)	*p* = 0.863
TC/CC	8 (100)	0 (0)		5 (63)	2 (25)	1 (12)	
TPMT:rs1800460							
CC	77 (91.6)	7 (8.3)	*p* = 1.0	47 (66)	14 (20)	10 (14)	*p* = 0.863
CT/TT	8 (100)	0 (0)		5 (63)	2 (25)	1 (12)	
COMT:rs4646316							
CC	50 (94)	3 (6)	*p* = 0.452	31 (66)	11 (23)	5 (11)	*p* = 0.459
CT/TT	35 (90)	4 (10)		21 (66)	5 (15)	6 (19)	
COMT:rs9332377							
CC	66 (93)	5 (7)	*p* = 0.657	40 (66)	11 (18)	10 (16)	*p* = 0.410
CT/TT	19 (90)	2 (10)		12 (67)	5 (28)	1 (5)	
ACYP2:rs1872328							
GG	79 (95)	4 (5)	*p =* 0.019 *	48 (67)	15 (21)	9 (12)	*p* = 0.499
GA	6 (67)	3 (33)		4 (57)	1 (14)	2 (29)	
WFS1:rs62283056							
GG	54 (89)	7 (11)	*p* = 0.091	35 (66)	12 (23)	6 (11)	*p =* 0.600
GC/CC	31 (100)	0 (0)	*p* = 0.091	17 (66)	4 (15)	5 (19)	

**Table 4 cancers-11-00551-t004:** Significance levels for genetic variants related to nephrotoxicity. * = significantly different (*p* < 0.05).

Genotype	No nephrotoxicity During Treatment *N* (%)	Nephrotoxicity during Treatment *N* (%)	*p*-Value	No Nephrotoxicity at End of Treatment *N* (%)	Nephrotoxicity at End of Treatment *N* (%)	*p*-Value
OCT2/SLC22A2:rs316019						
GG	35 (48.6)	37 (51.4)	*p* = 0.049*	61 (91.0)	6 (9.0)	*p* = 1.00
GT/TT	5 (24.0)	16 (76.0)		18 (89.5)	2 (10.5)	
MATE1:rs2289669						
GG	15 (42.9)	20 (57.1)	*p* = 1.00	29 (90.6)	3 (9.4)	*p* = 1.00
GA/AA	25 (43.1)	33 (56.9)		50 (90.9)	5 (9.1)	
XPD/ERCC2:rs13181						
TT	20 (55.6)	16 (44.4)	*p* = 0.085	32 (96.9)	1 (3.1)	*p* = 0.146
TG/GG	20 (35.7)	36 (64.3)		46 (86.8)	7 (13.2)	

**Table 5 cancers-11-00551-t005:** Overview of performed studies to ototoxicity and cisplatin.

Study	Discovery Xu [14]	Replication Xu [14]	Vos [13]	Thiesen [16]	Drogemoller [15]	Our Study
**Patients**	Children with brain tumours	Children with brain tumours	Children (3–43 yrs) with osteosarcoma	Children with different tumours	Testicular cancer	Head and neck cancer
**Number of patients**	238	68	156	149	229	92
**Cummulative dose cisplatin (median, range)**	287 mg/m^2^ (unknown)	Unknown *	480 mg/m^2^ (140–720)	378 mg/m^2^ (60–800)	400 mg/m^2^ (200–920)	240 mg/m^2^ (80–300)
**Concomitant drugs**	Vincristine, amisfostine, cyclofosfamide	Vinblastin, carboplatin	Vincristine, carboplatin in some pts	Vincristine, carboplatin	Etoposide, bleomycine	-
**Radiation**	Craniospinal	Focal in some pts	0	Some pts	0	IMRT

***** But same cisplatin dose as discovery cohort. IMRT: Intensity-modulated radiation therapy.

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
