# Peer review of "Genetic Variants as Predictive Markers for Ototoxicity and Nephrotoxicity in Patients with Locally Advanced Head and Neck Cancer Treated with Cisplatin-Containing Chemoradiotherapy (The PRONE Study)"

_cancers, 2019, doi:10.3390/cancers11040551_

Round 1

Reviewer 1 Report

I reviewed the manuscript by Driessen  et al., entitled ‘’Genetic variants as predictive markers for ototoxicity  and nephrotoxicity in patients with locally advanced  head and neck cancer treated with cisplatin containing chemoradiotherapy (the PRONE study)’’

The study is interesting, but needs some improvements. For example:

1.       The title is so long

2.       In the abstract the sentence ‘’ treated with cisplatin-containing chemoradiotherapy  ‘’ is not correct. It is a combination therapy in the form of radiation therapy and chemotherapy/cisplatin.  Accordingly the authors are asked to re-edit this sentence .

Many thanks

Author Response

We would like to thank the reviewer for the comments.

-  The title is so long

--> Answer: we think that this title comprise all the essential aspects of the study. but in case a shorter title is really wanted, we can agree with the following title:

Genetic variants for toxicity in head and neck cancer patients treated with cisplatin

- In the abstract the sentence ‘’ treated with cisplatin-containing chemoradiotherapy  ‘’ is not correct. It is a combination therapy in the form of radiation therapy and chemotherapy/cisplatin.  Accordingly the authors are asked to re-edit this sentence .

--> answer: We have replaced "cisplatin-containing chemoradiotherapy" by "chemoradiotherapy with cisplatin".

Reviewer 2 Report

This is a remarkable study evaluating the relationship between several genetic variants and cisplatin-induced ototoxicity and nephrotoxicity in locally advanced head and neck cancer (HNC). In particular, in this prospective observational study the Authors investigated 10 different single nucleotide polymorphisms in 92 patients, showing a significant association of the ACYP2 and OCT2 variants with ototoxicity and nephrotoxicity, respectively.

The techniques utilized were appropriate and described with plenty details. This is a well-designed study with rigorous methods. The discussion is well-balanced, and the statements are supported by the data.

I suggest adding some considerations in Discussion section (page 9, line 281):

“These data could be of particular interest, as several studies demonstrated the ability to enhance chemosensitivity to cisplatin in HNC, thus reducing the dosage needed, through the down-regulation of molecules related to cell death [1, 2].”

1.         Khan, Z., et al., Growth inhibition and chemo-radiosensitization of head and neck squamous cell carcinoma (HNSCC) by survivin-siRNA lentivirus. Radiother Oncol, 2016. 118(2): p. 359-68.

2.         Santarelli, A., et al., Nuclear Survivin as a Prognostic Factor in Squamous-Cell Carcinoma of the Oral Cavity. Appl Immunohistochem Mol Morphol, 2017. 25(8): p. 566-570.

Author Response

Thank you for the comments on our manuscript.

We added the suggestion and literatur references as mentioned by the reviewer.

Reviewer 3 Report

The authors conducted the prospective study with the cohort of 92 patient of LAHNC to investigate predictive markers for cisplatin-induced ototoxicity and nephrotoxicity. They confirmed that A allele for rs1872328 in the ACYP2 gene was significantly associated with cisplatin-induced clinical hearing loss.

As the authors mentioned, ACYP2 variant is already known to be associated cisplatin-induced hearing loss in other types of malignancy with larger sample sizes. 

Why was audiometrical hearing loss not associated with the variant of ACYP2? The reason of the discrepancy between clinical (subjective) and audiometrical (objective) ototoxicity should be discussed. Usually, objective data should be more trustworthy.

Figure 1 is really similar to previous paper's one (Ref 15). They added their own data, but it seems almost same.

In Table 1, the numbers of the patients may be not in correct positions (Treatment indication).

In Table 2, the numbers of the patients may be not in correct positions (Ototoxicity clinically at the end of treatment).

Author Response

We would like to thank the reviewer for his/her comments.

-Why was audiometrical hearing loss not associated with the variant of ACYP2? The reason of the discrepancy between clinical (subjective) and audiometrical (objective) ototoxicity should be discussed. Usually, objective data should be more trustworthy.

-->Answer: The reason for the discrepancy between clinical and audiometrical determined ototoxicity is speculative. Audiometrical determined ototoxicity is obviously more objective, but clinical assessed ototoxicity is probably more relevant for the patient.

We added this to the discussion section.

-Figure 1 is really similar to previous paper's one (Ref 15). They added their own data, but it seems almost same.

-->Answer: It is true that our figure is almost the same compared with the figure of Drogemoller et al., but this is what we expected to be, because we had a relatively small patient group and we found the same association as Drogemoller, so it would be unexpected if the tow figures differed a lot. But it illustrates that despite the fact that we could not find a significant difference between audiometrical determined ototoxicity and ACYP2, it strengthed the effect.

-In Table 1, the numbers of the patients may be not in correct positions (Treatment indication).

-In Table 2, the numbers of the patients may be not in correct positions (Ototoxicity clinically at the end of treatment).

--> Answer: We changed the format of the tables according to the MDPI style.

Round 2

Reviewer 3 Report

The clinical (subjective) ototoxicity can not exclude possibility of psychological effects by chemotherapy. A allele of ACYP2 could be associated with suggestible personality. The eudiometrical (objective) exam was supposed to exclude it, but they had a discrepancy. This is the most important issue for the scientific impact of this report. The integrity of the paper is high, and the authors' results will be true. But, I have to say that this paper is not scientific enough for the journal with impact factor over 5.0.